# Habitual exercise evokes fast and persistent adaptation during split-belt walking

**Sarah A. Brinkerhoff** [1] *, **Natalia Sánchez** [2‡], **Jaimie A. Roper** [1‡]

**1** School of Kinesiology, Auburn University, Auburn, Alabama, United States of America, **2** Department of Physical Therapy, Chapman University, Irvine, California, United States of America

‡ NS and JAR are joint senior authors on this work
* sbrinkerhoff@auburn.edu

**Data Availability Statement:** All data files and analyses are available from the Figshare database (https://doi.org/10.6084/m9.figshare.c.6607117.v2).

## Abstract

Changing movement patterns in response to environmental perturbations is a critical aspect of gait and is related to reducing the energetic cost of the movement. Exercise improves energetic capacity for submaximal exercise and may affect how people adapt movement to reach an energetic minimum. The purpose of this study was to determine whether self-reported exercise behavior influences gait adaptation in young adults. Young adults who met the optimal volume of exercise according to the Physical Activity Guidelines for Americans (MOVE; n = 19) and young adults who did not meet the optimal volume of exercise (notMOVE; n = 13) walked on a split-belt treadmill with one belt moving twice the speed of the other belt for 10 minutes. Step length asymmetry (SLA) and mechanical work done by each leg were measured. Nonlinear mixed effects models compared the time course of adaptation between MOVE and notMOVE, and t-tests compared net work at the end of adaptation between MOVE and notMOVE. Compared to notMOVE, MOVE had a faster initial response to the split belt treadmill, and continued to adapt over the duration of split-belt treadmill walking. Young adults who engage in sufficient amounts of exercise responded more quickly to the onset of a perturbation, and throughout the perturbation they continued to explore movement strategies, which might be related to reduction of energetic cost. Our findings provide insights into the multisystem positive effects of exercise, including walking adaptation.

## Introduction

Adapting walking patterns in response to environmental perturbations is a critical aspect of locomotion. When a person encounters a perturbation, such as when walking on an icy or uneven surface, they must adapt their walking patterns to avoid falling which can be achieved using different strategies. While prior research suggests that visual feedback [1, 2], focus of attention [3, 4], and neurological injury [5–7] can affect aspects of walking adaptation, how or if individual factors related to overall physical activity might influence walking adaptation strategies needs further evidence.

A common approach to study walking adaptation is split-belt treadmill walking, in which the belts under each leg move at different speeds [8, 9]. The changes in the asymmetry between

**Funding:** This work was supported by the Auburn University College of Education under Seed Grant [JR18SG to J.A.R.]. The funders had no role in study design, data collection and analysis, decision to publish, or preparation of the manuscript.

**Competing interests:** The authors have declared that no competing interests exist.

left and right step lengths, or step length asymmetry (SLA) is one measure used to track how a person's gait pattern adapts in response to a continuous perturbation. As a robust measure of gait adaptation, SLA is observable with the unaided eye, is sensitive to experimental manipulations, and persists even after the split-belt perturbation is removed [1–4, 10]. In line with upper extremity motor adaptation [11], adaptation of SLA during split-belt walking occurs at two distinct and interacting timescales–a fast component that adapts rapidly and a slow component that adapts more gradually [12–15]. From work in both upper and lower extremity motor tasks, the two timescales of motor adaptation may derive from two separate (but not necessarily independent) processes.

Indeed, Seethapathi and colleagues posited that the component that adapts rapidly is driven by balance optimization, whereas the component that adapts more gradually is driven by energetic cost optimization [16]. Consistent with these findings, experimental studies have shown that SLA adaptation during split-belt walking occurs in parallel with reductions in the work generated by the legs; gait adaptation results in decreasing positive work and increasing negative work done by the legs, especially the leg on the fast belt [17, 18]. As the legs reduce positive work done and increase negative work done, energetic cost decreases concomitantly [17, 19, 20]. This complements the fact that doing negative work is less energetically costly than doing positive work [21, 22]. By reducing work done by the legs, people are likely gradually adapting towards some lower energetic cost, as determined by Seethapathi et al. [16] and previously empirically suggested [23–25]. Similarly, a study by Park et al. showed that during early adaptation to split belt walking, individuals increase whole body angular momentum, which is inversely proportional to balance [26]. Therefore, these experimental findings also support the idea of balance and energetic cost as the two distinct processes driving gait adaptation.

The question then arises—do individual factors that affect energy consumption affect the ability to adapt gait during split-belt walking? The multisystem benefits of exercise are well-known and include improved physical function, cognition, quality of life, and reduced risk of cardiovascular disease and all-cause mortality [27]. Moreover, the ability to achieve minimum energetic cost of transport while running is contingent upon a person's level of aerobic training experience. People who engage in more aerobic training are able to reach this optimal cost of transport, whereas people with less aerobic training are not [28]. Habitual physical activity is generally known to improve aerobic capacity [29, 30]. Exercise-induced adaptations would increase capacity for exercise and enable trained individuals to tolerate submaximal exercise for longer. In this study, we examine two competing hypotheses regarding the adaptation to split-belt walking and its impact on energetic cost. The first hypothesis suggests that individuals who engage in more exercise would reach an energetic optimum faster, aiming to reduce energetic cost [28]. Conversely, the second hypothesis suggests that individuals who engage in more exercise may more gradually approach an energetic optimum due to their greater tolerance for submaximal exercise [31, 32]. These two hypotheses indicate that exercise habits might influence adaptation toward more energetically economical movement patterns.

The purpose of this study was to determine whether self-reported exercise behavior influences gait adaptation in young adults. We hypothesized that amount of self-reported exercise would affect gait adaptation, and this effect would primarily be driven by differences in the slow component of adaptation, given the role of energetics in shaping the rate of adaptation of the slow component [16, 23–25] and the well-established effects of exercise on energetics [28, 31, 32]. Our hypothesis is based on two opposing mechanisms which we will test here: 1) Young adults engaging in sufficient weekly exercise (Meets Optimal Volume of Exercise; MOVE) would adapt faster than those engaging in low or no weekly exercise (does not Meet Optimal Volume of Exercise; notMOVE) to reach a minimum energetic cost quickly [28], evidenced by faster reduction in SLA and work done by the legs; 2) MOVE would have a higher

tolerance for the submaximal exercise of treadmill walking than notMOVE, evidenced by more gradual reduction in SLA and work done by the legs. To test these competing mechanisms, we analyzed the adaptation of SLA and the positive and negative work done by each leg while MOVE and notMOVE walked on a split-belt treadmill. We also hypothesized that at the end of adaptation, MOVE would perform more-negative net work than notMOVE, by reducing positive work done by the legs and increasing negative work done by the legs in order to reduce overall energetic cost [17, 19, 20]. A significant effect of regular exercise participation on gait adaptation would support the idea that individual factors that influence energetics can also influence walking adaptation. Our findings will provide insights into the multisystem positive effects of exercise, including walking adaptation, and will provide directions for exercise rehabilitation research.

## Methods

### Participants

We recruited a convenience sample of young adults ages 19–35 for participation in this study. Participants were excluded if they reported cardiovascular, pulmonary, renal, metabolic, vestibular, or neurologic disorders; any lower-extremity injuries or surgeries in the past 12 months; or a prior anterior cruciate ligament injury. The Auburn University Institutional Review Board approved all procedures and all participants completed written informed consent prior to participating (Protocol #18–418 MR 1811). Data were collected in the Locomotor and Movement Control Lab at Auburn University.

### Experimental protocol

Participants first completed a self-report exercise behavior questionnaire that consisted of a modified Godin-Leisure Time Exercise Questionnaire [33] and a custom survey on sport and exercise modalities. This questionnaire asked how many times per week participants currently exercised at a moderate-to-vigorous intensity, for how many minutes per session, and for how many consecutive weeks. Based on their responses, participants were grouped into one of two groups according to The Physical Activity Guidelines for Americans: MOVE (at least 150 minutes per week of exercise for at least the last three months) or notMOVE (less than 150 minutes per week of exercise for at least the last three months) [27]. The questionnaire also asked what types of exercise participants were currently engaged in. For this study, we excluded participants who engaged only in weight training or flexibility exercise, in an attempt to bias the sample towards aerobic-based activities.

After completing the questionnaire, participants walked on the treadmill. Participants held onto side handrails for the entirety of treadmill walking. The instructions given to participants noted that they should use the handrails for stability, and not to offload bodyweight. First, we determined participants' typical and fastest comfortable walking speeds on the treadmill using a modified protocol by Dingwell and Marin [34]. Starting at 0.6 m/s, the treadmill incrementally increased by 0.05 m/s every 4 seconds until participants reported that the current speed was their typical walking speed, or was their fastest comfortable speed. This was repeated twice for typical walking, and twice for fast walking. We instructed participants either, "tell me when you reach your typical walking speed," or, "tell me when you reach the fastest speed that you'd be comfortable walking for ten minutes." The fastest comfortable speed was set as the fast belt velocity, and the slow belt velocity was set as half of the fastest comfortable speed.

The split-belt walking protocol is shown in Fig 1. After finding their typical and fast speeds, participants warmed up to the treadmill and were familiarized to the belt speeds first by walking for three minutes at their typical walking speed, second by walking for three minutes at

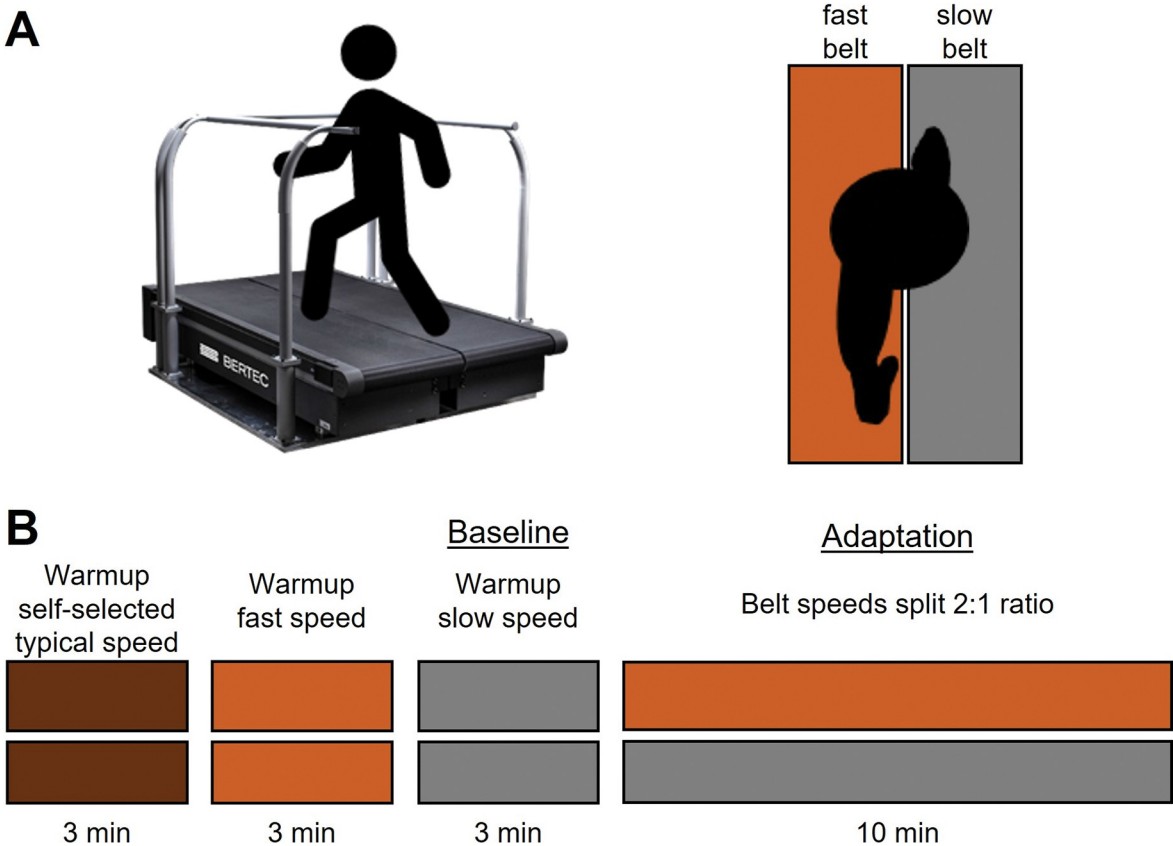

**Fig 1. Split-belt walking protocol.** A) An oblique and birds-eye view of the split-belt treadmill. B) First, participants warmed up to the treadmill and were familiarized to the belt speeds first by walking for three minutes at their typical walking speed, second by walking for three minutes at the fast speed, and third by walking for three minutes at the slow speed. Finally, they walked for 10 minutes with the belt under their dominant leg at the fast speed, and the belt under their non-dominant leg at the slow speed.

their fastest comfortable speed, and third by walking for three minutes at half of the fast speed (slow speed). The last tied walking condition—three minutes of walking at the slow speed—was considered the baseline condition for data analysis. We did not randomize the order of the warmup trials, to ensure that the baseline condition was the same across participants. Then, participants walked for ten minutes with the belt speeds split—the belt under the dominant leg moved at participants' fastest comfortable speed, and the belt under the non-dominant leg moved at the slow speed. Leg dominance was determined as the leg that a participant reported they would use to kick a ball.

## Data analysis

Kinematic data were recorded at 100 Hz from reflective markers placed bilaterally on the anterior superior iliac spine and the lateral malleoli of the ankles, using a 17-camera motion capture system (VICON; Vicon Motion Systems Ltd, Oxford, United Kingdom). Kinematic data were lowpass filtered with a 4th order Butterworth filter with a cutoff frequency of 6 Hz. Ground reaction force data were obtained for each individual leg using an instrumented split-belt treadmill, recorded at 1000 Hz from two separate force plates. Force data were lowpass filtered with a 4th order Butterworth filter with a cutoff frequency of 20 Hz.

Step length was calculated as the distance between the ankle markers along the anterior-posterior walking axis at foot strike. SLA was calculated and normalized to stride length as in

Eq 1.

$$Step\ length\ Asymmetry = \frac{Step\ length_{fast} - Step\ length_{slow}}{Step\ length_{fast} + Step\ length_{slow}} \qquad 1$$

Here, $Step\ Length_{fast}$ is the step length when the leg on the fast belt strikes the belt, and $Step\ Length_{slow}$ is the step length when the leg on the slow belt strikes the belt. A negative SLA indicates that the leg on the slow belt is taking a longer step than the leg on the fast belt, and an SLA of zero indicates that the legs are taking steps of equal length.

We calculated positive and negative mechanical work generated by the legs across each stride cycle using a custom MATLAB program. A stride cycle was calculated as the time between ipsilateral foot-strikes. We used the point-mass model to estimate mechanical work generated by the legs on the treadmill and on the center of mass [15, 17, 18]. In brief, we segmented force data into strides [18] and calculated the center of mass velocities in each direction as the time integral of the center of mass accelerations. Next, we calculated the instantaneous power generated by each leg for each stride as the instantaneous sum of the dot product of the ground reaction force and the center of mass velocity and the dot product of the force applied to the respective belt and the belt speed. We then calculated the total positive and total negative work performed by each leg as the time integral of the positive or negative portion of the total instantaneous power over the stride cycle. We calculated work rate by dividing each work measure by stride duration. Finally, we calculated the net work rate by the legs at the end of adaptation as the time integral of the power divided by stride cycle, averaged over the last 100 strides.

## Statistical analysis

All statistical analyses were conducted in R [35]. We used Welch Two Sample t-tests to compare age, height, leg length, mass, and fast-leg belt speed between groups. We also used Pearson's Chi-squared tests with Yates' continuity correction to assess the difference in proportion of males and females in each group, and to assess the difference in proportion of people who walked with the right leg on the fast belt in each group. We analyzed SLA, positive and negative work rate by the fast legs, and positive and negative work rate by the slow leg using mixed effects nonlinear regression models with the 'nlme' package [36]. We truncated the data for all participants to minimum number of steps taken by a participant (823 steps) so that all participants would weight equally on each step.

Detailed description of the statistical models can be found in the supporting information. In brief, for each of the work rate outcome measures and for SLA, we built-up models to two-exponent models—assuming that adaptation of the outcome measures occurred over two timescales—that included a fixed effect for group (Eqs 2 and 3). Here, $c$ is the estimated plateau if strides went to infinity; $a_f$ is the initial value of the fast component of adaptation; $r_f$ is the growth rate of the fast component of adaptation; $a_s$ is the initial value of the slow component of adaptation; $r_s$ is the growth rate of the slow component of adaptation. In the models that included a fixed effect for group, all five model parameters ($c$, $a_f$, $r_f$, $a_s$, $r_s$) were allowed to differ for MOVE vs. notMOVE.

$$SLA = \left( c + a_f * e^{-\frac{step}{r_f}} + a_s * e^{-\frac{step}{r_s}} \right) \sim Group + (c|ID) \qquad 2$$

$$Work\ rate = \left( c + a_f * e^{-\frac{stride}{r_f}} + a_s * e^{-\frac{stride}{r_s}} \right) \sim Group + (c|ID) \qquad 3$$

We compared these two-exponent models with fixed effects for group to models without fixed effects for group, and to one-exponent models assuming adaptation occurs over a single timescale. All models and outcomes can be found in the supporting information. All models were fitted with a maximum likelihood estimation and all models contained a random effect such that the outcome measure's estimated plateau was allowed to vary by participant. The simplest, best-fitting model for each measure was the final model that we interpreted, such that a lower Akaike Information Criterion (AIC) and Bayesian Information Criteria (BIC) indicated a better fit to the data and log-likelihood tests assessed goodness of fit to the data [37, 38]. If the final model included a fixed effect for group, the significance (or lack thereof) of the effect of group on the equation variables in the best-fitting models was evaluated using Welch's t-test with an *a priori* alpha level of 0.05. If the final model included a fixed effect for group, initial adaptation values were calculated for each variable as the average over the first five strides and were compared between group using Welch's Two Sample t-test.

We used Welch's Two Sample t-test between groups to compare the net work rate by the legs at the end of adaptation, to determine if there was a difference between exercise groups in the ability to gain assistance from the treadmill.

## Results

### Participants

Thirty-seven people participated in this study. We excluded five participants who engaged in weightlifting, yoga, and scuba diving, in an attempt to bias the sample towards aerobic-based activities. Therefore, 32 participants were included in these analyses (Table 1). Based on the self-report exercise questionnaire, 19 participants were placed in the MOVE group and 13 were placed in the notMOVE group. The two groups did not significantly differ by sex, height, mass, leg on the fast belt, or fast belt speed, but did significantly differ by age where the MOVE group was slightly older. Demographic and belt speed data are provided in Table 1.

### General timescales of adaptation

The best-fitting model for each outcome measure was the two-exponent model containing a fixed effect for group (Eqs 2 and 3, Table 2), indicating that SLA and work rate by the legs adapted over two distinct timescales and was affected by exercise. All candidate models for each outcome measure, and the final model fits to each participant's data, can be found in the supporting information. AIC, BIC, and log-likelihood tests for goodness of fit agreed for all outcome measures.

**Table 1. Participant demographics in mean [range].**

|  | notMOVE | MOVE | *p-value* | Total |
|---:|---|---|---|---|
| *n* | *13* | *19* | — | *32* |
| Age (yr) | 20 [19,22] | 22 [19,32] | **0.045** | 22 [19,32] |
| Height (cm) | 1706 [1540,1910] | 1727 [1540,1920] | 0.565 | 1718 [1540,1920] |
| Mass (kg) | 66.5 [48,89] | 72.9 [50,116] | 0.208 | 70.3 [48,116] |
| Fast belt speed (m/s) | 1.57 [1.25,1.95] | 1.49 [1.15,1.98] | 0.231 | 1.52 [1.15,1.98] |
| Minutes/week exercise | 38 [0,120] | 268 [150,750] | $<$**0.001** | 175 [0,750] |
| Sex | 11 female, 2 male | 11 female, 8 male | 0.225 | 22 female, 10 male |
| Fast leg | 3 right, 10 left | 3 right, 16 left | 0.954 | 6 right, 26 left |
| Exercise modality | run, cycle, basketball, tennis | run, cycle, basketball, tennis, CrossFit®, dance, boxing | — | — |

*Note*. Type of exercise participated in reflects the exercise modalities of people who participated in any exercise every week (regardless of minutes per week of exercise).

**Table 2. Final mixed effects models for SLA, positive and negative work of the fast leg, and positive and negative work of the slow leg.**

| | Coefficient | SLA | Positive Work Rate | | Negative Work Rate | |
|---|---|---|---|---|---|---|
| | | | Fast Leg | Slow Leg | Fast Leg | Slow Leg |
| notMOVE | $c$ (plateau) | -0.043 *** (0.010) | 0.54 *** (0.12) | 0.20 (0.17) | -0.28 ** (0.10) | -0.48 ** (0.16) |
| | $a_f$ (fast component initial value) | -0.097 *** (0.007) | 0.38 *** (0.06) | 0.28 *** (0.03) | 0.02 (0.02) | -0.60 *** (0.06) |
| | $r_f$ (fast component growth rate) | 32.465 *** (3.751) | 5.19 *** (1.47) | 7.56 *** (1.33) | 31.94 (48.41) | 6.26 *** (1.18) |
| | $a_s$ (slow component initial value) | -0.017 *** (0.008) | 0.66 ** (0.03) | 0.32 * (0.15) | 0.03 (0.04) | -0.66 *** (0.03) |
| | $r_s$ (slow component growth rate) | 162.122 ** (69.910) | 86.45 *** (5.92) | 629.28 (423.13) | 1.12 (3.53) | 101.41 *** (8.35) |
| MOVE | $c$ (plateau) | -0.033 (0.014) $p = 0.475$ | 0.68 (0.16) $p = 0.359$ | 0.36 (0.18) $p = 0.362$ | -0.24 (0.14) $p = 0.754$ | -0.68 (0.21) $p = 0.341$ |
| | $a_f$ (fast component initial value) | -0.078 [†] (0.009) $p = 0.032$ | 0.56 [†] (0.07) $p = 0.009$ | 0.55 [†] (0.04) $P<0.001$ | -0.15 [†] (0.04) $p<0.001$ | -0.831 [†] (0.07) $p = 0.002$ |
| | $r_f$ (fast component growth rate) | 11.628 [†] (3.929) $p<0.001$ | 6.78 (1.74) $p = 0.362$ | 3.45 [†] (1.36) $p = 0.003$ | 1.86 (48.41) $p = 0.534$ | 6.76 (1.38) $p = 0.718$ |
| | $a_s$ (slow component initial value) | -0.060 [†] (0.008) $p<0.001$ | 0.36 [†] (0.03) $p<0.001$ | 0.05 (0.15) $p = 0.064$ | 0.12 (0.07) $p = 0.194$ | -0.40 [†] (0.03) $p<0.001$ |
| | $r_s$ (slow component growth rate) | 396.341 [†] (78.306) $p = 0.003$ | 157.47 [†] (23.30) $p = 0.002$ | 321.16 (579.39) $p = 0.595$ | 433.6 (304.94) $p = 0.157$ | 156.07 [†] (23.47) $p = 0.20$ |
| | AIC | -106168.7 | 2021.38 | -12838.49 | -12350.91 | 4848.64 |
| | BIC | -106070.6 | 2111.25 | -12748.6 | -12261.04 | 4938.51 |
| | Log-likelihood | 53096.36 | -998.69 | 6431.24 | 6187.46 | -2412.32 |
| | Number of observations | 26336 | 13216 | 13216 | 13216 | 13216 |
| | Number of participants | 32 | 32 | 32 | 32 | 32 |
| | SD: ID ($c$) | 0.038 | 0.435 | 0.231 | 0.347 | 0.572 |
| | SD: Residual ($c$) | 0.032 | 0.259 | 0.147 | 0.150 | 0.288 |

*Note*. Model parameters are given as Coefficient (SEM), *p*-value. The best-fitting model to the data for all five outcome measures was the two-exponent model with a fixed effect on all variables by group. All models contain a random plateau on participant. AIC = Akaike Information Criterion; BIC = Bayesian Information Criterion. Values in parentheses are standard errors of the fixed effects. Asterisks indicate significant effects where

*p<0.05

**p<0.01

***p<0.001.

Daggers on coefficients in the MOVE group indicate coefficients that are significantly different between MOVE and notMOVE, and all group-difference p-values are shown.

## Step length asymmetry

Compared to notMOVE, MOVE adapted SLA more gradually. The groups differed in every model-estimated coefficient except for SLA plateau. Overall, MOVE and notMOVE did not differ in initial SLA (MOVE = -0.144, notMOVE = -0.137, $t(28.763) = 0.283$, $p = 0.779$)). MOVE adapted the fast component of SLA more quickly than notMOVE (MOVE $r_f = 12$ steps, notMOVE $r_f = 32$ steps), seen in the quicker achievement of the "elbow" by MOVE in Fig 2. Conversely, MOVE adapted the slow component of SLA more slowly than notMOVE (MOVE $r_s = 396$ steps, notMOVE $r_s = 162$ steps), seen in the gradual change in SLA after the elbow in Fig 2. Notably, the variability in SLA at each step between subjects appeared much larger across MOVE than across notMOVE (Fig 2). For better understanding of SLA adaptation, Fig 2B shows the averaged individual step lengths for fast and slow legs.

The model explained some of the variance in SLA plateau (Table 2, SD: Residual ($c$) = 0.032). Variation between participants accounted for a large portion of the variance in SLA plateau (Table 2, SD: ID ($c$) = 0.038).

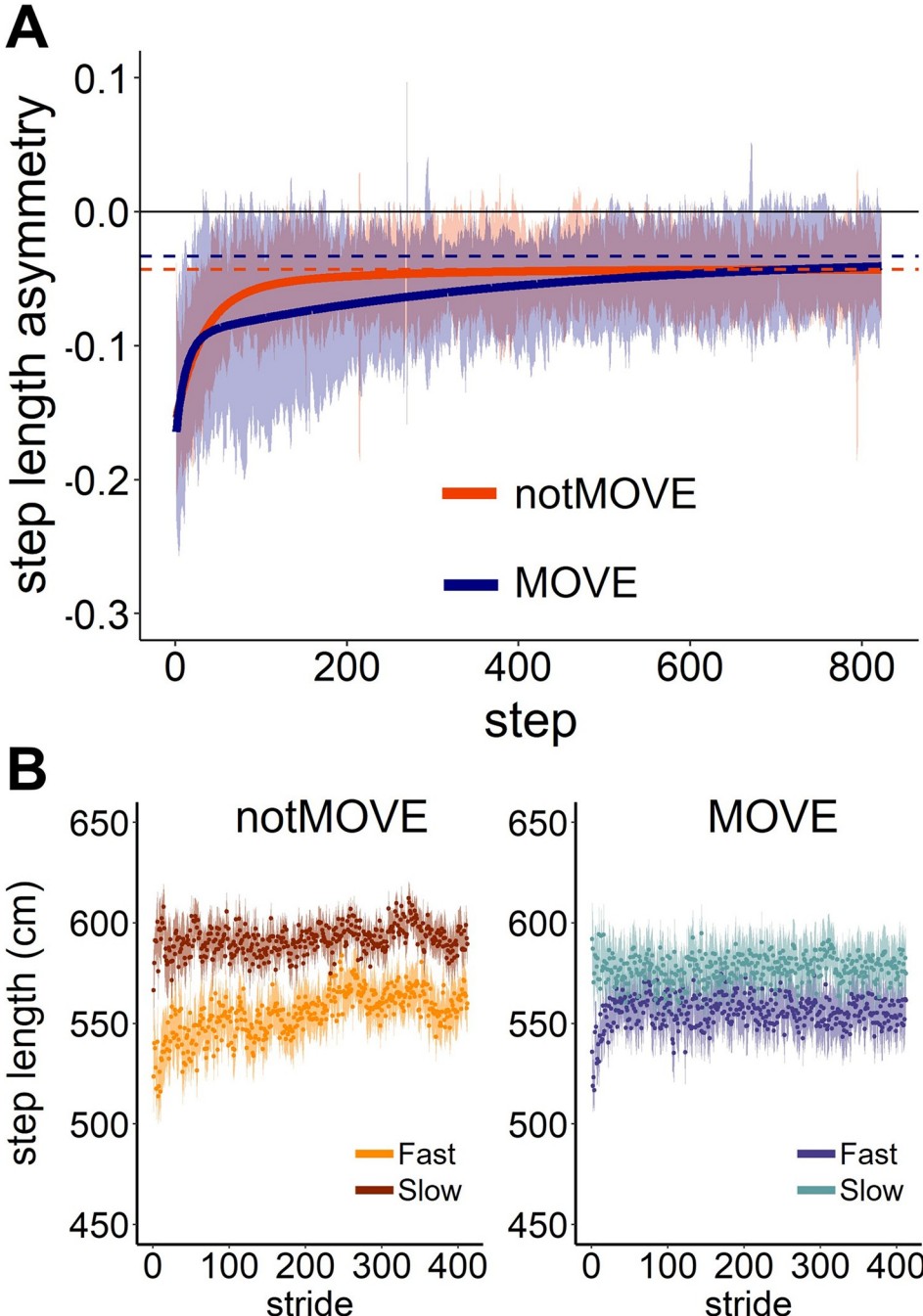

**Fig 2. Step length asymmetry adapts differently over 823 steps of split-belt treadmill walking.** (A) Step length asymmetry adaptation. Shaded areas indicate the standard deviation. Solid lines indicate the model fit to the data. Dashed lines indicate the model-estimated plateau for each group. Blue = MOVE ($n$ = 19), orange = notMOVE ($n$ = 13). (B) Average step lengths for the fast and slow legs during adaptation, for each group.

## Positive work rate by the fast leg

Compared to notMOVE, MOVE continued to gradually adapt positive work rate by the fast leg over the entire trial. The groups differed in the initial values and the slow-component growth rate (Fig 3A). Overall, MOVE and notMOVE did not differ in initial positive work rate

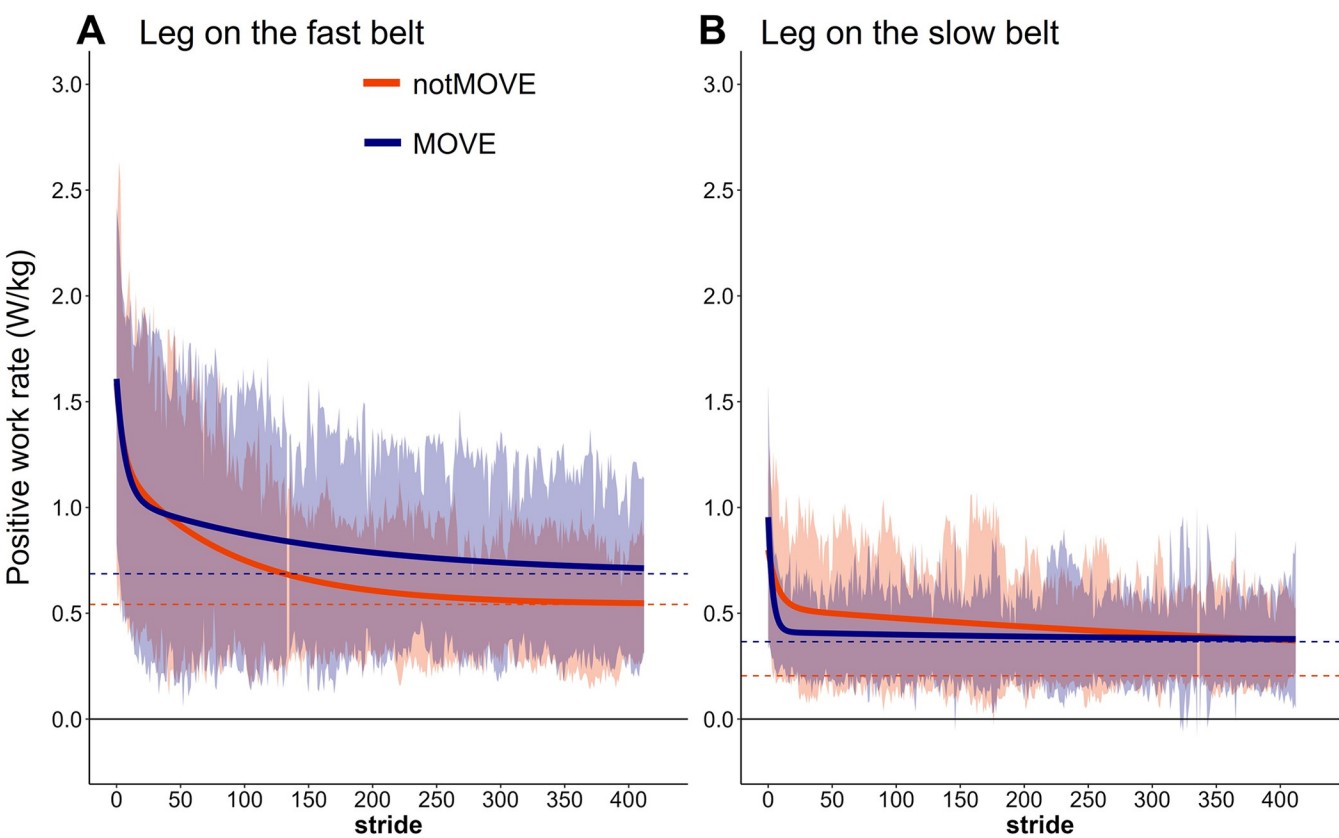

**Fig 3.** Positive work rate by the legs on the fast (A) and slow (B) belts over 823 steps of split-belt treadmill walking. Shaded areas indicate the standard deviation of each group. Solid lines indicate the model fit to the data. Dashed lines indicate the model-estimated plateau for each group. Blue = MOVE ($n$ = 19), orange = notMOVE ($n$ = 13).

by the fast leg (MOVE = 1.44 W/kg, notMOVE = 1.45 W/kg, $t$(23.484) = 0.051, $p$ = 0.960). MOVE adapted the slow component of positive work rate by the fast leg more slowly than not-MOVE did (Fig 3A) (MOVE $r_s$ = 157 strides, notMOVE $r_s$ = 86 strides).

The model explained much of the variance in fast-leg positive work rate plateau (Table 2, SD: Residual ($c$) = 0.259). Variation between participants accounted for a large portion of the variance in fast-leg positive work rate plateau (Table 2, SD: ID ($c$) = 0.435).

The variability in SLA in positive work by the fast leg appeared higher in the MOVE group than in the notMOVE group (Figs 2 and 3A). To test this, we conducted follow-up analyses of the within-participant SLA and positive fast-leg work rate variability [39]. We calculated the standard deviation of each measure in the initial 100 strides, middle 100 strides, and final 100 strides. Separate student's two-sample t-tests compared variability of each measure during the initial, middle, and final epochs of gait adaptation between the active and inactive groups. There was no difference between groups in within-participant variability in either measure during any epoch.

### Negative work rate by the fast leg

MOVE and notMOVE did not differ in adaptation of the negative work rate by the fast leg. There was no significant adaptation in negative work rate by the fast leg, in either group (Fig 4A).

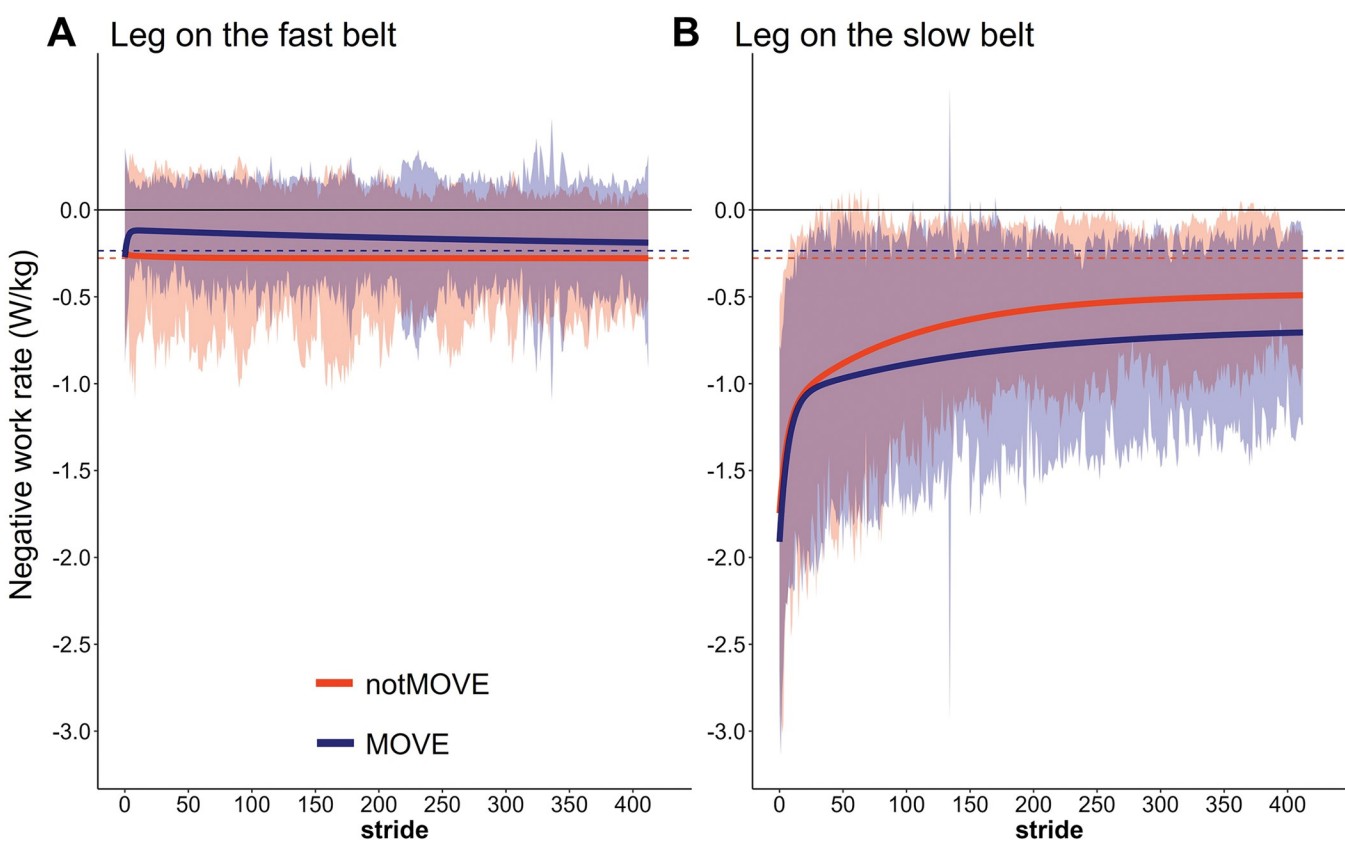

**Fig 4.** Negative work rate by the legs on the fast (A) and slow (B) belts over 823 steps of split-belt treadmill walking. Shaded areas indicate the standard deviation of each group. Solid lines indicate the model fit to the data. Dashed lines indicate the model-estimated plateau for each group. Blue = MOVE ($n$ = 19), orange = notMOVE ($n$ = 13).

The model explained much of the variance in fast-leg negative work rate plateau (Table 2, SD: Residual ($c$) = 0.150). Variation between participants accounted for a large portion of the variance in fast-leg negative work rate plateau (Table 2, SD: ID ($c$) = 0.347).

### Positive work rate by the slow leg

Overall, MOVE and notMOVE did not differ in initial positive work rate by the slow leg (MOVE = 0.701 W/kg, notMOVE = 0.722 W/kg, $t$(23.462) = 0.147, $p$ = 0.885). MOVE adapted the fast component of positive work rate by the slow leg more quickly than notMOVE did (Fig 3B) (MOVE $r_f$ = 3 strides, notMOVE $r_f$ = 8 strides). Both groups converged on a plateau by about 10 strides, and the groups did not differ in their estimated plateaus.

The model explained much of the variance in slow-leg positive work rate plateau (Table 2, SD: Residual ($c$) = 0.147). Variation between participants accounted for a large portion of the variance in slow-leg positive work rate plateau (Table 2, SD: ID ($c$) = 0.231).

### Negative work rate by the slow leg

Compared to notMOVE, MOVE adapted the slow component of negative work rate by the slow leg more gradually (Fig 4B). Overall, MOVE and notMOVE did not differ in initial negative work rate by the slow leg (MOVE = -1.664 W/kg, notMOVE = -1.57 W/kg, $t$(23.717) = 0.248, $p$ = 0.806). MOVE adapted the slow component significantly slower (MOVE $r_s$ = 156 strides, notMOVE $r_s$ = 101 strides). The groups did not differ in their estimated plateaus.

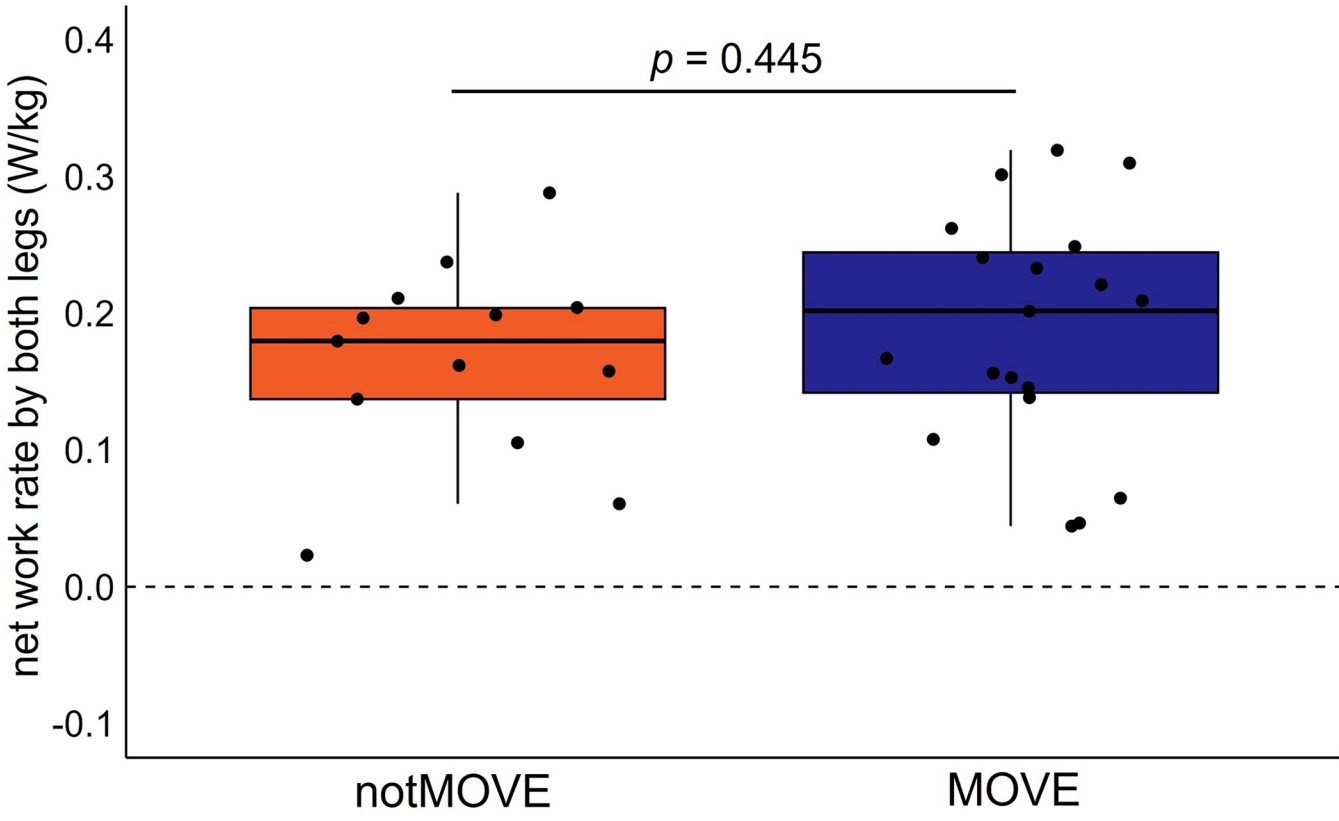

**Fig 5. Net work rate by the legs.** Blue = MOVE ($n$ = 19), Orange = notMOVE ($n$ = 13).

The model explained much of the variance in slow-leg negative work rate plateau (Table 2, SD: Residual ($c$) = 0.288). Variation between participants accounted for a large portion of the variance in slow-leg negative work rate plateau (Table 2, SD: ID ($c$) = 0.572).

### Net work rate

The groups did not differ in the total net work rate by the legs ($t$(28.549) = -0.774, $p$ = 0.445) (Fig 5). MOVE performed 0.166 (SD 0.072) W/kg and notMOVE performed 0.188 (SD 0.085) W/kg at the end of ten minutes of split-belt walking.

### Discussion

The purpose of this study was to determine whether self-reported exercise behavior influences gait adaptation in young adults. Our main findings are: 1) Compared to notMOVE, MOVE more gradually adapted the slow timescale of SLA, positive work rate by the fast leg, and negative work rate by the slow leg over the entirety of split-belt exposure, 2) MOVE initially adapted the fast timescale of SLA and positive work rate by the slow leg quicker than notMOVE, 3) Neither group adapted the negative work rate by the fast leg, and 4) There was no effect of exercise on net work done by the legs. The findings from this study support the hypothesis that young adults who engage in sufficient amounts of weekly exercise have a higher tolerance for the energetically-challenging asymmetric belt speeds and adapt more gradually than those who do not.

MOVE continued to gradually adapt over the entirety of split-belt exposure. These current findings complement the literature. If the gradual component of gait adaptation is driven by energetic cost optimization [16], then long-term involvement in regular exercise—affecting exercise capacity—would affect the gradual component of gait adaptation, as seen here in both SLA adaptation and in the reduction in the positive work rate by the fast leg and negative work rate by the slow leg. Exercise-trained adults have an enhanced exercise capacity relative to non-exercisers, evidenced by higher maximal oxygen uptake (VO2max) [31]; a better exercise economy [31, 32]; and higher lactate and ventilatory thresholds [31]. A higher exercise capacity indicates that MOVE can tolerate submaximal exercise—such as treadmill walking—at a given energetic cost for longer than notMOVE, consistent with our results. Our results suggest that young adults, regardless of exercise habits, can reduce positive work by the fast leg during split-belt adaptation, but that the amount of exercise practiced weekly and regularly affects the rate at which positive work rate reduction occurs. Those engaging in suboptimal amounts of exercise (notMOVE) may have a direr need to reduce positive work by the leg than those who engage in sufficient amounts of exercise (MOVE), because they may be working at a higher relative effort, resulting in a stronger incentive to reduce energetic and therefore mechanical work [15, 17]. Our results contradict the potential mechanism that MOVE would be quicker at reducing energetic cost than notMOVE. While adults who engage in more exercise per week do reach a minimum cost of transport during running, unlike those engaging in less exercise [28], our results suggest that adults who engage in more exercise per week do not have an advantage in reaching a minimum energetic cost during split-belt adaptation. Therefore, high amounts of regular weekly exercise may allow people to have a higher tolerance for the perturbation induced by split-belt walking, allowing them to adapt their gait more gradually.

The MOVE group initially adapted SLA and positive work rate by the slow leg quicker than the notMOVE group. Our findings suggest that young adults who engage in more exercise are quickly able to modify their gait in response to a continuous perturbation, possibly improving stability [16] more quickly than those who engage in less exercise. Indeed, athletes perform better than non-athletes on a balance task after a learning block of multimodal balance training [40]. Buurke and colleagues investigated the relationship between mediolateral stability adaptation and metabolic cost adaptation during 9 minutes of 2:1 split-belt walking [23]. However, they did not find a relationship between mediolateral stability and the reduction in metabolic cost. While their finding seems in contrast to both our findings and the theory that initial rapid gait adaptations are intended to optimize balance [16], all these findings taken together support a nuanced approach to motor skill learning. One explanation, supported by the two-timescale model, is that adults prioritize balance over metabolic cost when first perturbed, to meet the goals of the task (here, the assumed goal being to stay upright and continue walking). Then, once balance is achieved, metabolic cost becomes more relevant.

Neither group adapted the negative work rate by the fast leg. Negative work rate by the fast leg during split-belt adaptation was largely unaffected by exercise. Previous work reported that over forty-five minutes of adaptation at a 3:1 belt speed ratio, negative work rate by the fast leg increased prior to positive work rate by the fast leg decreasing [15]. Our finding of no adaptation of negative work rate by the fast leg, by either group contrasts our hypothesis that exercise capacity affects metabolic adaptation through increasing the negative work rate done by the legs. A short adaptation period at a smaller belt speed ratio coupled with high variability across participants could explain the lack of significant change in the negative work rate by the fast leg in the current study, compared to previous work [15].

Both the MOVE and notMOVE groups performed net positive work by the end of ten minutes of gait adaptation. Sánchez and colleagues reported that, at positive values of SLA—achieved by enough time spent split-belt walking [15]—the legs perform net negative work

[17]. This suggests that people are able to reduce overall positive work and increase overall negative work performed. In contrast, our sample performed net positive, not negative, work. The difference in findings between studies is likely due to the shorter adaptation period (ten versus forty-five minutes) and the smaller split-belt speed ratio (2:1 Vs. 3:1) in the current study, leading to negative average values of SLA at the end of adaptation. Given enough time and/or a larger belt speed ratio, our sample would likely have plateaued at positive SLA values and performed net negative work. Future research should explore if exercise affects the ability to perform net negative work over prolonged adaptation periods at more extreme split-belt perturbations.

Variability in SLA and positive work by the fast leg appeared larger in the MOVE group than in the notMOVE group. Abrams and colleagues determined that during 45 minutes of 3:1 split-belt treadmill walking, young adults reduced the variability in SLA over the course of split-belt treadmill walking [39]. If the MOVE group had a larger tolerance for submaximal exercise than did the notMOVE group, then they may also have had a larger window of tolerance for movement variability during initial adaptation. However, our results do not support the theory that adults who exercise more would employ a more varied hypothesis testing strategy during early adaptation during ten minutes of split-belt treadmill walking. The larger variation in the standard error of the MOVE group in SLA and positive fast-leg work rate is therefore likely because the individual responses to the perturbation within the MOVE group is varied, while individual responses in the notMOVE group were more homogenous. Further individual factors that can explain the variation in active individuals' response should be investigated.

There are some limitations to this study. First, participants held on to the handrails during treadmill walking, which likely influenced both balance and mechanical work calculations. While handrail holding was kept constant across participants and across groups, future studies should investigate the effect of exercise on gait adaptation without holding on to handrails. Second, we did not directly measure metabolic energetic cost using expired gas analyses. It is possible that the MOVE group may have modified SLA and mechanical work rate through nonmetabolic processes, such as tendon stiffness, without modifying metabolic cost. Our groupings based on self-reported exercise training and not including resistance-trained adults reasonably allows us to infer that reported differences between groups would be due to differences in energetic capacity. Future studies examining the metabolic adaptations between those who meet the exercise recommendations and those who do not is a warranted next-step. Third, we standardized belt speeds to participants' fastest comfortable walking speeds, which did not differ between groups. This method of selecting the fast-belt speed has been used previously [41–43], and because there was not a statistical difference between groups in initial SLA, we know that the groups were not perturbed differently relative to their baseline, despite different belt speeds. Fourth, we used self-report measures of exercise participation. Habitual exercise recall has a moderate validity correlation to actual exercise (0.36) [44] and, despite its drawbacks, is the most common way to assess physical activity [45–47]. Additionally, the presence of group differences in the current study suggests that self-report exercise habits were keen enough to separate participants by energetic capacity that would affect gait adaptation, though it is possible that participants practiced other exercise modalities that they did not disclose. Finally, we did not test retention of the adapted gait pattern, so we cannot comment on whether exercise affects young adults' magnitude or persistence of errors during gait pattern washout.

The amount of self-reported habitual exercise affects a young adult's response to a novel perturbation. Our results suggest that people who regularly participate in sufficient exercise respond more quickly to the onset of a perturbation and continue to explore strategies and

search for assistance from the environment in order to reduce energetic cost. The effect of exercise training on movement adaptation likely acts through increasing aerobic capacity, thus reducing relative effort of a given submaximal movement. On a split-belt treadmill, MOVE may require longer time than notMOVE to fully take advantage of the work done by the treadmill; however, requiring a longer time to adapt is not inherently unfavorable. In fact, if provided enough time to fine-tune their response to the perturbation, MOVE may plateau at a more-positive SLA, a more-negative net work done by the legs, and therefore perhaps a lower energetic cost than notMOVE. The current results provide preliminary evidence that MOVE may be more able than notMOVE to use the work done by the treadmill to adapt and reduce positive work done by the legs, given enough time.

## Supporting information

**S1 File. Supplemental statistical methods.**
(PDF)

**S1 Table. Model fits of mixed effects linear models for each outcome measure.** *Note*: All models contain a random plateau on participant. Model 1 contains fixed effects for the parameters in the one-exponent equation (Eq 1), fitted to the whole sample. Model 2 contains fixed effects for the parameters in the one-exponent equation, fitted by group. Model 3 contains fixed effects for the parameters in the two-exponent equation (Eq 2), fitted to the whole sample. Model 4 contains fixed effects for the parameters in the two-exponent equation, fitted by group. AIC = Akaike Information Criterion; BIC = Bayesian Information Criterion. The final model for each outcome measure is in bold (model 4 for all outcome measures).
(PDF)

**S1 Fig. A comparison of the group average model fit (black line) and the individual model fits (blue line) of step length asymmetry adaptation.**
(PDF)

**S2 Fig. A comparison of the group average model fit (black line) and the individual model fits (blue line) of the adaptation of the positive work rate of the fast leg.**
(PDF)

**S3 Fig. A comparison of the group average model fit (black line) and the individual model fits (blue line) of the adaptation of the negative work rate of the fast leg.**
(PDF)

**S4 Fig. A comparison of the group average model fit (black line) and the individual model fits (blue line) of the adaptation of the positive work rate of the slow leg.**
(PDF)

**S5 Fig. A comparison of the group average model fit (black line) and the individual model fits (blue line) of the adaptation of the positive work rate of the slow leg.**
(PDF)

## Acknowledgments

We thank the individuals who participated in the study and the undergraduate research assistants who assisted with data collection and processing. We also thank Maxwell Donelan, PhD and Surabhi Simha PhD, for the MATLAB code on which we based our code to calculate positive and negative work generated by the legs.

## Author Contributions

**Conceptualization:** Sarah A. Brinkerhoff, Natalia Sánchez, Jaimie A. Roper.

**Data curation:** Sarah A. Brinkerhoff.

**Formal analysis:** Sarah A. Brinkerhoff.

**Funding acquisition:** Jaimie A. Roper.

**Methodology:** Sarah A. Brinkerhoff.

**Resources:** Natalia Sánchez.

**Supervision:** Natalia Sánchez, Jaimie A. Roper.

**Visualization:** Sarah A. Brinkerhoff.

**Writing – original draft:** Sarah A. Brinkerhoff, Natalia Sánchez, Jaimie A. Roper.

**Writing – review & editing:** Sarah A. Brinkerhoff, Natalia Sánchez, Jaimie A. Roper.

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
