## [Decision Letter · Decision Letter 0]

2 Mar 2023

PONE-D-23-01404

Habitual aerobic exercise evokes fast and persistent adaptation during split-belt walking

PLOS ONE

Dear Dr. Sarah A Brinkerhoff,

Thank you for submitting your manuscript to PLOS ONE. After careful consideration, we feel that it has merit but does not fully meet PLOS ONE’s publication criteria as it currently stands. Therefore, we invite you to submit a revised version of the manuscript that addresses the points raised during the review process.

<ul> <li> 

 <li> 

 <li> 

We look forward to receiving your revised manuscript.

Kind regards,

Flávio Oliveira Pires, PhD

Academic Editor

PLOS ONE

Journal Requirements:

"This work was supported by the Auburn University College of Education under Seed Grant [JR18SG to J.A.R.]."

"This work was supported by the Auburn University College of Education under Seed Grant [JR18SG to J.A.R.]. The funders had no role in study design, data collection and analysis, decision to publish, or preparation of the manuscript."

3. We note that you have stated that you will provide repository information for your data at acceptance. Should your manuscript be accepted for publication, we will hold it until you provide the relevant accession numbers or DOIs necessary to access your data. If you wish to make changes to your Data Availability statement, please describe these changes in your cover letter and we will update your Data Availability statement to reflect the information you provid

Additional Editor Comments:

Please, special attention to reviewer's comments on the open data and analysis, please follow PLosOne's policy.

Reviewers' comments:

Reviewer's Responses to Questions

**Comments to the Author**

1. Is the manuscript technically sound, and do the data support the conclusions?

Reviewer #1: Partly

Reviewer #2: Yes

2. Has the statistical analysis been performed appropriately and rigorously? 

Reviewer #1: Yes

Reviewer #2: Yes

3. Have the authors made all data underlying the findings in their manuscript fully available?

Reviewer #1: No

Reviewer #2: Yes

4. Is the manuscript presented in an intelligible fashion and written in standard English?

Reviewer #1: Yes

Reviewer #2: Yes

5. Review Comments to the Author

Reviewer #1:

Brinkerhoff and colleagues present data from an experiment in which two groups of young adults -- habitual exercisers and non-exercisers -- adapted to split-belt perturbations in walking. They measured step-length asymmetry as well as several work-rate measures to contrast opposing hypotheses: Exercisers adapting more quickly (because they are more skilled at finding an optimal gait pattern) or slowly (because they can tolerate sub-optimal movement better). Modelling adaptation as a combination of fast and slow processes, they find stronger fast-process adaptation in non-exercisers across outcome measures, consistent with the second hypothesis.

This is an interesting and well-written manuscript. The experimental design is straightforward and suitable to answering a very relevant research question. However, I have a few questions and comments regarding the modelling and analyses.

Please find my specific comments below.

1. I would ask the authors to provide their data and analyses on a public repository, currently I cannot find them.

2. I can see why the authors would let participants choose a comfortable speed as the speed of the fast belt, but this seems like a potentially confounding variable. Did the authors assess whether faster belt speeds were related to the outcome variables, especially SLA?

3. I am trying to reconcile the main finding that non exercisers' fast component of SLA adaptation was higher with figure 2B. Especially in fig 2B, it looks like if anything, exercisers adapted faster. However, the data for exercisers appeared to be a lot more variable. Seeing the distributions of parameter values (as well as potentially the correlations between them) would go a long way towards being able to interpret these mean differences. This also highlights the importance of making data and analyses available.

4. Perhaps I missed it, why use steps for modelling SLA but strides for work rate?

5. The models are not overly complex and there is no word limit for the methods section, so I do not see a good reason to put them in the supplementary information.

6. l.210 "ΔAIC > 2 indicated a better fit to the data (31,32)." I am not sure this is correct as stated. Burnham and Anderson consider a difference of 2 or more to be the threshold for substantial support for a model, but in general, the model with the lower AIC (even if the difference is smaller than 2) is the better fitting model. This is true regardless of the number of parameters, as AIC already includes a summand of 2*K with K being the number of parameters.

7. On a related note, did the authors use AIC or BIC to decide which model to use -- or was there never disagreement between the two?

8. l. 226 The height should be in cm, not mm.

9. Table 1 is a bit hard to read, perhaps the authors should decide to use one of sd and range (I think either is fine).

Reviewer #2:

Overall / Pg --- / Line --- / Comment: In general, the authors have done an excellent job, showing great dedication and care for the paper. I believe that some changes may be necessary to clarify aspects related to the purpose of the study, as well as a more precise definition of the different groups.

Title / Pg 1 / Line 1 / Comment: I suggest reviewing the use of the word "aerobic". In the next comments, the reason for the suggestion will become clearer.

Abstract / Pg 2 / Line 27-28 / Comment: The purpose of the study suggests that differences in recreational aerobic exercise will affect gait adaptation. However, as the participants were not exposed to interventions, it seems more appropriate to indicate that the possible differences are related to the amount of physical activity (greater or lesser than 150 min/week).

Abstract / Pg 2 / Line 32-35 / Comment: I believe that the repetition of the terms Habitual Exercisers and Non-exercisers could be reduced somewhat. Perhaps consider the use of acronyms in the text as a whole.

Introduction / Pg 3 / Line 43-44 / Comment: A more faithful example of everyday life could be given instead of "walking on boat rocking on the water".

Introduction / Pg 3 / Line 45-46 / Comment: The sentence: "How individual factors might influence walking adaptation strategies remains to be determined", seems loose and disconnected. I suggest including (with references) the potential factors that may influence walking adaptation strategies, and then highlighting that individual factors still lack further evidence.

Introduction / Pg 3 / Line 50 / Comment: Adaptations to what?

Introduction / Pg 3 / Line 50-52 / Comment: The sentence: "SLA robustly changes during split-belt walking, shows an aftereffect after the split-belt perturbation is removed, is observable with the unaided eye, and is sensitive to experimental manipulations", seems confuse. I suggest reformulating the sentence paying attention to the following points: 1) Verb tense used in the word "shows"; 2) Term "aftereffect after"; 3) Better connection in the complement sentence "is observable with the unaided eye".

Introduction / Pg 3 / Line 53 and 55 / Comment: The terms "two distinct rates" and "two timescales" represent the same concept? If yes, why not standardized?

Introduction / Pg 4 / Line 71-87 / Comment: Despite finding the debate extremely relevant, it seems to me that there is an overload of information about aerobic exercise that may not be consistent with the study's method, since, the amount of physical activity of the participants was measured from responses to a questionnaire. Therefore, a question remains: Is this questionnaire able to clearly identify that in the self-reported weekly volume of physical activities, only aerobic exercises are included? I am afraid that other activities of a different nature may be included in participant's responses. In this way, it is possible that the weekly volume reported includes strength and flexibility exercises, etc.

Introduction / Pg 4 / Line 88-89 / Comment: The purpose of the study was more adequately described here compared to the abstract.

Introduction / Pg 4 / Line 89-90 / Comment: Here, your hypothesis seems more appropriate based on the amount of exercise and not on aerobic exercise. I also suggest reviewing the title of the study.

Methods - Participants / Pg 5 / Line 109 / Comment: Any kind of a priori calculation was done to determine the sample size? Or a posteriori to determine the power?

Methods - Experimental Protocol / Pg 6 / Line 119-121 / Comment: Which questionnaire was used to determine the amount of physical activity? Please include references.

Methods - Experimental Protocol / Pg 6 / Line 122-125 / Comment: The classification of participants was based on the American College of Sports Medicine or the Physical Activity Guidelines for Americans? Despite the recommendation being the same, I believe that they are different documents and only one of them (REF 24) was cited.

Methods - Experimental Protocol / Pg 6 / Line 125-127 / Comment: I believe that this information can be relocated just above, in the part where the questionnaire is announced. Again, the reference is missing.

Methods - Experimental Protocol / Pg 6 / Line 132-139 / Comment: I felt that this part of the text is quite truncated. Particularly with many repetitions of the word "speed".

Methods - Experimental Protocol / Pg 6 / Line 133 / Comment: I suggest clearly stating what "relatively slow speed" means in km/h or m/h.

Methods - Experimental Protocol / Pg 6 / Line 141-143 / Comment: The order of application of walking speeds (typical, comfortable fastest and slow) was randomized? If so, please state clearly in the text, including that this is a warm up. This is highlighted in the figure, but not in the text. If not, please highlight whether this protocol may have been influenced by the order effect.

Methods - Statistical Analysis / Pg 9 / Line 199 / Comment: Is the equation that appears on line 167 not included in the equation count?

Results / Pg 10 / Line 229-232 / Comment: I believe that the way in which the results are being presented can be standardized. At the beginning of the sentence, the authors talk about the variables that did not show a difference and chose to show the mean values of only one of them. Still at the end, t and p values of variables that showed difference are brought. Perhaps table 1 could contain this t and p information for all variables with no need to repeat in the text.

Table 1 / Pg 10 / Line 235 / Comment: It seems contradictory to me to call the participants "NON-exercises" and say that they practice running, cycling, basketball and tennis. It may be necessary to revise the term in the text as a whole.

Discussion / Pg 16 / Line 343-344 / Comment: The purpose of the study highlighted in the abstract, introduction and now in the discussion are presenting different objectives between them. I suggest reviewing.

Discussion / Pg 16 / Line 349-351 / Comment: The sentence: "young adults who habitually exercise have a higher tolerance for the energetically-challenging asymmetric belt speeds and adapt more gradually than THOSE WHO DO NOT", seems imprecise since according to table 1, all individuals practiced exercise. The difference between them is in the amount. This term "who do not exercise" appears at other times in the text, for example in line 376. I suggest reviewing the entire text.

Discussion / Pg 16 / Line 357-366 / Comment: I assume that all information provided in this excerpt is indeed relevant. However, it seems to me that a more adequate path could draw a parallel between the amount of physical activity practiced weekly and the potential influences that this generates on exercise capacity, VO2, gait, etc. Although what is written is true, the way it is, it implies that the participants of the NON-EXERCISE group do not practice any activity, which was not demonstrated by table 1. I suggest reviewing this and other excerpts that deal with the theme in this way , so that the differences between trained, insufficiently trained and untrained individuals can be clearer.

Discussion / Pg 16-17 / Line 367-370 / Comment: Truncated sentence, I suggest rephrasing.

Discussion / Pg 17 / Line 376-377 / Comment: I'm not sure this would be the best evidence for the moment. This paper (REF 33) talks about athletes and static balance.

Discussion / Pg 18 / Line 397-398 / Comment: Why not use the acronym SLA?

6. PLOS authors have the option to publish the peer review history of their article (what does this mean?). If published, this will include your full peer review and any attached files.

Reviewer #1: No

Reviewer #2: No

---

## [Author Response · Author response to Decision Letter 0]

26 Apr 2023

Journal Requirements:

Author response:

We updated the formatting of the manuscript and title page to adhere to the PLOS ONE guidelines.

Thank you for stating the following in the Acknowledgments Section of your manuscript: "This work was supported by the Auburn University College of Education under Seed Grant [JR18SG to J.A.R.]." We note that you have provided funding information that is not currently declared in your Funding Statement. However, funding information should not appear in the Acknowledgments section or other areas of your manuscript. We will only publish funding information present in the Funding Statement section of the online submission form. Please remove any funding-related text from the manuscript and let us know how you would like to update your Funding Statement. Currently, your Funding Statement reads as follows: "This work was supported by the Auburn University College of Education under Seed Grant [JR18SG to J.A.R.]. The funders had no role in study design, data collection and analysis, decision to publish, or preparation of the manuscript." Please include your amended statements within your cover letter; we will change the online submission form on your behalf.

Author response:

We have removed funding-related text from the manuscript and we have included the correct funding statement in the cover letter and here:

“This work was supported by the Auburn University College of Education under Seed Grant [JR18SG to J.A.R.]. The funders had no role in study design, data collection and analysis, decision to publish, or preparation of the manuscript."

We note that you have stated that you will provide repository information for your data at acceptance. Should your manuscript be accepted for publication, we will hold it until you provide the relevant accession numbers or DOIs necessary to access your data. If you wish to make changes to your Data Availability statement, please describe these changes in your cover letter and we will update your Data Availability statement to reflect the information you provide.

Author response:

We have uploaded our data and the SLA analysis to FigShare, and have made them publicly available. https://doi.org/10.6084/m9.figshare.c.6607117.v2

Additional Editor Comments:

Please, special attention to reviewer's comments on the open data and analysis, please follow PLosOne's policy.

Author response:

We have uploaded our data and the SLA analysis to FigShare, and have made them publicly available. https://doi.org/10.6084/m9.figshare.c.6607117.v2

 

Reviewer 1:

Brinkerhoff and colleagues present data from an experiment in which two groups of young adults -- habitual exercisers and non-exercisers -- adapted to split-belt perturbations in walking. They measured step-length asymmetry as well as several work-rate measures to contrast opposing hypotheses: Exercisers adapting more quickly (because they are more skilled at finding an optimal gait pattern) or slowly (because they can tolerate sub-optimal movement better). Modelling adaptation as a combination of fast and slow processes, they find stronger fast-process adaptation in non-exercisers across outcome measures, consistent with the second hypothesis.

This is an interesting and well-written manuscript. The experimental design is straightforward and suitable to answering a very relevant research question. However, I have a few questions and comments regarding the modelling and analyses.

Author response:

The authors thank the reviewer for their time and energy put into reviewing our work. One main change we made in response to the other reviewer was to change the group designations to Meets Optimal Volume of Exercise; MOVE (formerly Habitual Exercisers) and does not Meet Optimal Volume of Exercise; notMOVE (formerly Non-exercisers).

1. I would ask the authors to provide their data and analyses on a public repository, currently I cannot find them.

Author response:

We have uploaded our data and the SLA analysis to FigShare, and have made them publicly available. https://doi.org/10.6084/m9.figshare.c.6607117.v

2. I can see why the authors would let participants choose a comfortable speed as the speed of the fast belt, but this seems like a potentially confounding variable. Did the authors assess whether faster belt speeds were related to the outcome variables, especially SLA?

Author response:

Before completing our primary analysis, we ran a preliminary Pearson correlation between fast-belt speed and SLA at the end of the adaptation. We found that belt speed was not related to the magnitude of end-adaptation SLA (r=0.071, p=0.701). We also assessed whether the groups chose significantly different fast-belt speeds, and they did not (MOVE = 1.49 m/s, notMOVE= 1.57 m/s; t(29.617)=1.224, p=0.231). Therefore, we moved forward with the proposed analysis without controlling belt speeds.

3. I am trying to reconcile the main finding that non exercisers' fast component of SLA adaptation was higher with figure 2B. Especially in fig 2B, it looks like if anything, exercisers adapted faster. However, the data for exercisers appeared to be a lot more variable. Seeing the distributions of parameter values (as well as potentially the correlations between them) would go a long way towards being able to interpret these mean differences. This also highlights the importance of making data and analyses available.

Author response:

We have updated the table note for Table 2 to include the following:

Line 277: “Model parameters are given as Coefficient (SEM), p-value.”

We have also made the data and analyses publicly available on FigShare, https://doi.org/10.6084/m9.figshare.c.6607117.v2. Also, we ran a follow-up analysis comparing within-subject variability across groups over the course of split-belt walking. We found that there was no difference in within-subject variability across groups, which would suggest that the individual responses to the perturbation within the MOVE group is more varied, while individual responses in the notMOVE group were more homogenous. This analysis and discussion have bene added to the discussion section, starting at line 418.

4. Perhaps I missed it, why use steps for modelling SLA but strides for work rate?

Author response:

We used the most granular longitudinal variable that made sense for our outcome measures. Step length is defined as the distance between the ankles at each heel-strike. This allows us to then calculate asymmetry between the fast and slow legs (figure 1). Step length is measured for strides, too, as a stride requires both steps. Conversely, the definition of positive and negative mechanical work is the” time integral of the positive or negative portion of the total instantaneous power over the stride cycle.” We have added terminology to indicate that the work rate is not a “length” measure, but a temporal value.

Line 188-189: “A stride cycle was calculated as the time between ipsilateral foot-strikes.”

5. The models are not overly complex and there is no word limit for the methods section, so I do not see a good reason to put them in the supplementary information.

Author response:

Thank you for your valuable feedback on our manuscript. We appreciate your input and have carefully considered your suggestion regarding the placement of the methods section. After discussion, we have decided to keep the methods section in the supplementary information. We believe that this approach will help maintain the clarity and conciseness of the main message in the manuscript, while still providing detailed technical information for those who are interested. Thank you again for your time and thoughtful comments.

6. l.210 "ΔAIC > 2 indicated a better fit to the data (31,32)." I am not sure this is correct as stated. Burnham and Anderson consider a difference of 2 or more to be the threshold for substantial support for a model, but in general, the model with the lower AIC (even if the difference is smaller than 2) is the better fitting model. This is true regardless of the number of parameters, as AIC already includes a summand of 2*K with K being the number of parameters.

Author response:

We removed the “requirement” of an AIC difference greater than or equal to 2 to determine best model fit. It also should be noted that the differences between the selected models and the next-best-fitting models were considerably greater than 2 (smallest AIC difference = 65.62, Negative work rate of the fast leg)

7. On a related note, did the authors use AIC or BIC to decide which model to use -- or was there never disagreement between the two?

Author response:

AIC and BIC model comparison agreed for all outcome measures. This statement has been added to the text.

Line 258-259: “AIC, BIC, and log-likelihood tests for goodness of fit agreed for all outcome measures.”

8. l. 226 The height should be in cm, not mm.

Author response:

We corrected this error in both the text and Table 1.

9. Table 1 is a bit hard to read, perhaps the authors should decide to use one of sd and range (I think either is fine).

Author response:

We updated Table 1 and chose to report means and ranges. We also reorganized the table for better clarity.

Reviewer 2:

Overall / Pg --- / Line --- / Comment: In general, the authors have done an excellent job, showing great dedication and care for the paper. I believe that some changes may be necessary to clarify aspects related to the purpose of the study, as well as a more precise definition of the different groups.

Author response:

We would like to thank the reviewer for the time and effort put into their review. As overall responses, first, we revised and made consistent all the purpose statement in the manuscript. Second, we updated the group designations to Meets Optimal Volume of Exercise; MOVE (formerly Habitual Exercisers) and does not Meet Optimal Volume of Exercise; notMOVE (formerly Non-exercisers).

Title / Pg 1 / Line 1 / Comment: I suggest reviewing the use of the word "aerobic". In the next comments, the reason for the suggestion will become clearer.

Author response:

We added information on study exclusion to the methods:

Line 241-243: “Thirty-seven people participated in this study. We excluded five participants who engaged in weightlifting, yoga, and scuba diving, in an attempt to bias the sample towards aerobic-based activities. Therefore, 32 participants were included in these analyses (Table 1).”

Abstract / Pg 2 / Line 27-28 / Comment: The purpose of the study suggests that differences in recreational aerobic exercise will affect gait adaptation. However, as the participants were not exposed to interventions, it seems more appropriate to indicate that the possible differences are related to the amount of physical activity (greater or lesser than 150 min/week).

Author response:

We updated the group designations to MOVE and notMOVE groups. We have clarified the methods, too, to specify that we included only participants who engaged in some modality of exercise with an aerobic component (in both the High and low Exercise groups). Finally, in regards to this comment, we have updated the purpose (in the abstract, introduction, and discussion) to state:

“The purpose of this study was to determine whether self-reported exercise behavior influences gait adaptation in young adults.”

Abstract / Pg 2 / Line 32-35 / Comment: I believe that the repetition of the terms Habitual Exercisers and Non-exercisers could be reduced somewhat. Perhaps consider the use of acronyms in the text as a whole.

Author response:

We updated the group designations to MOVE and notMOVE groups.

Introduction / Pg 3 / Line 43-44 / Comment: A more faithful example of everyday life could be given instead of "walking on boat rocking on the water".

Author response:

We have updated the examples given to read as follows:

Line 51-53: “When a person encounters a perturbation, such as when walking on an icy or uneven surface, they must adapt their walking patterns to avoid falling, which can be achieved using different strategies.”

Introduction / Pg 3 / Line 45-46 / Comment: The sentence: "How individual factors might influence walking adaptation strategies remains to be determined", seems loose and disconnected. I suggest including (with references) the potential factors that may influence walking adaptation strategies, and then highlighting that individual factors still lack further evidence.

Author response:

We added specific examples with references to the potential influencing factors of gait adaptation. 

Line 53-56: “While prior research suggests that visual feedback [1,2], focus of attention [3,4], and neurological injury [5–7] can affect aspects of walking adaptation, how or if individual factors related to overall physical activity might influence walking adaptation strategies needs further evidence.”

Introduction / Pg 3 / Line 50 / Comment: Adaptations to what?

Author response:

We revised the sentence to be more specific.

Line 59-60: “The changes in the asymmetry between left and right step lengths, or step length asymmetry (SLA) is one measure used to track how a person’s gait pattern adapts in response to a continuous perturbation.”

Introduction / Pg 3 / Line 50-52 / Comment: The sentence: "SLA robustly changes during split-belt walking, shows an aftereffect after the split-belt perturbation is removed, is observable with the unaided eye, and is sensitive to experimental manipulations", seems confuse. I suggest reformulating the sentence paying attention to the following points: 1) Verb tense used in the word "shows"; 2) Term "aftereffect after"; 3) Better connection in the complement sentence "is observable with the unaided eye".

Author response:

We rewrote the sentence for clarity, as the reviewer suggested.

Line : “As a robust measure of gait adaptation, SLA is observable with the unaided eye, is sensitive to experimental manipulations, and persists even after the split-belt perturbation is removed [1–4,10].”

Introduction / Pg 3 / Line 53 and 55 / Comment: The terms "two distinct rates" and "two timescales" represent the same concept? If yes, why not standardized?

Author response:

We revised the terms in the paper to consistently refer to the concept as “two timescales.” This included removing two instances of “dual rate” and “distinct rate.” The sentence the reviewer referred to now reads as follows:

Line 62-64: “In line with upper extremity motor adaptation (8), adaptation of SLA during split-belt walking occurs at two distinct and interacting timescales…”

Introduction / Pg 4 / Line 71-87 / Comment: Despite finding the debate extremely relevant, it seems to me that there is an overload of information about aerobic exercise that may not be consistent with the study's method, since, the amount of physical activity of the participants was measured from responses to a questionnaire. Therefore, a question remains: Is this questionnaire able to clearly identify that in the self-reported weekly volume of physical activities, only aerobic exercises are included? I am afraid that other activities of a different nature may be included in participant's responses. In this way, it is possible that the weekly volume reported includes strength and flexibility exercises, etc.

Author response:

The questionnaire asked participants to state the types of exercise that they were currently regularly engaged in. As we did not track participants outside of the lab, self-report data is the best way we have to understand the participant’s exercise habits. We added a statement to the limitations section:

Line 457-458: “Additionally, the presence of group differences in the current study suggests that self-report exercise habits were keen enough to separate participants by energetic capacity that would affect gait adaptation, though it is possible that participants practiced other exercise modalities that they did not disclose.”

For this study, we excluded participants who only engaged in weight lifting or flexibility training, and one participant who only engaged in a scuba diving class.

We updated the participants section of the results to indicate the exclusion of non-aerobic-exercising participants:

Line #: “Thirty-seven people participated in this study. We excluded five participants who singularly currently engaged in weightlifting, yoga, and a scuba diving course, in an attempt to homogenize the sample towards aerobic-based activities. Therefore, 32 participants were included in this analysis (Table 1).”

Introduction / Pg 4 / Line 88-89 / Comment: The purpose of the study was more adequately described here compared to the abstract.

Author response:

We have updated the abstract, introduction, and discussion to reflect the same purpose:

“The purpose of this study was to determine whether self-reported exercise behavior influences gait adaptation in young adults.”

Introduction / Pg 4 / Line 89-90 / Comment: Here, your hypothesis seems more appropriate based on the amount of exercise and not on aerobic exercise. I also suggest reviewing the title of the study.

Author response:

We have slightly revised this sentence to read,

Line 98-102: “We hypothesized that amount of self-reported exercise would affect gait adaptation, and this effect would primarily be driven by differences in the slow component of adaptation, given the role of energetics in shaping the rate of adaptation of the slow component [16,23–25] and the well-established effects of exercise on energetics [28,31,32].”

We have also revised the title to be: 

“Habitual exercise evokes fast and persistent adaptation during split-belt walking”

Methods - Participants / Pg 5 / Line 109 / Comment: Any kind of a priori calculation was done to determine the sample size? Or a posteriori to determine the power?

Author response:

This was a secondary data analysis born from a discussion at a conference. To ensure the study was powered, we conducted a postpriori power calculation for the final SLA model (two timescales and a fixed effect of group) determining the percent of 1000 bootstrapped samples with significant group effects for each timescales of adaptation (r1 and r2). While this power analysis is not exact, because our primary analysis compared model fits to the data for best fit, it is an estimate of the final model’s statistical power. The power analysis reported that 99% of the models contained significant group effects for r1, and 58% of models contained significant group effects for r2, indicating that the timescales of adaptation may be consistently different from zero in a majority of the bootstrapped samples.

Methods - Experimental Protocol / Pg 6 / Line 119-121 / Comment: Which questionnaire was used to determine the amount of physical activity? Please include references.

Author response:

We used a combination of the Godin-Leisure Time Exercise Questionnaire (modified to remove MET-minutes per week and only look at minutes per week) and a custom set of questions that asked participants to specify the types/modalities of mild/moderate/vigorous exercise that they engaged. The methods sentence has been updated as follows.

Line 129-131: “Participants first completed a self-report exercise behavior questionnaire that consisted of a modified Godin-Leisure Time Exercise Questionnaire [31] and a custom survey on sport and exercise modalities.”

Methods - Experimental Protocol / Pg 6 / Line 122-125 / Comment: The classification of participants was based on the American College of Sports Medicine or the Physical Activity Guidelines for Americans? Despite the recommendation being the same, I believe that they are different documents and only one of them (REF 24) was cited.

Author response:

The Physical Activity Guidelines for Americans is the correct document that we cited. We updated the sentence in the methods accordingly.

Line 134: “Based on their responses, participants were grouped into one of two groups according to The Physical Activity Guidelines for Americans…”

Methods - Experimental Protocol / Pg 6 / Line 125-127 / Comment: I believe that this information can be relocated just above, in the part where the questionnaire is announced. Again, the reference is missing.

Author response:

We apologize for the confusion on the questionnaire. Participants were separately asked which leg they kicked a ball with and this was not part of the exercise questionnaire but was a separate question asked by the researcher. We moved this sentence (and revised it) to the section when asymmetric belt speeds are first mentioned:

Line 161-162: “Leg dominance was determined as the leg that a participant reported they would use to kick a ball.”

Methods - Experimental Protocol / Pg 6 / Line 132-139 / Comment: I felt that this part of the text is quite truncated. Particularly with many repetitions of the word "speed".

Author response:

We revised this portion of the text in attempts to make it read more fluidly.

Line 144-151: “Starting at 0.6 m/s, the treadmill incrementally increased by 0.05 m/s every 4 seconds until participants reported that the current speed was their typical walking speed, or was their fastest comfortable speed. This was repeated twice for typical walking, and twice for fast walking. We instructed participants either, “tell me when you reach your typical walking speed,” or, “tell me when you reach the fastest speed that you’d be comfortable walking for ten minutes.” The fastest comfortable speed was set as the fast belt velocity, and the slow belt velocity was set as half of the fastest comfortable speed.”

Methods - Experimental Protocol / Pg 6 / Line 133 / Comment: I suggest clearly stating what "relatively slow speed" means in km/h or m/h.

Author response:

We revised this statement to include the exact speed-finding protocol.

Line 144-156: “Starting at 0.6 m/s, the treadmill incrementally increased by 0.05 m/s every 4 seconds until participants reported that the current speed was their typical walking speed, or was their fastest comfortable speed.”

Methods - Experimental Protocol / Pg 6 / Line 141-143 / Comment: The order of application of walking speeds (typical, comfortable fastest and slow) was randomized? If so, please state clearly in the text, including that this is a warm up. This is highlighted in the figure, but not in the text. If not, please highlight whether this protocol may have been influenced by the order effect.

Author response:

The order of the 3 warmup trials (typical, fast, and slow speeds) was not randomized. 

Line 152-158: “After finding their typical and fast speeds, participants warmed up to the treadmill and were familiarized to the belt speeds first by walking for three minutes at their typical walking speed, second by walking for three minutes at their fastest comfortable speed, and third by walking for three minutes at half of the fast speed (slow speed ). The last tied walking condition — three minutes of walking at the slow speed — was considered the baseline condition for data analysis. We did not randomize the order of the warmup trials, to ensure that the baseline condition was the same across participants.”

Methods - Statistical Analysis / Pg 9 / Line 199 / Comment: Is the equation that appears on line 167 not included in the equation count?

Author response:

We updated all equation numbers accordingly, to include the SLA equation as eq. 1.

Results / Pg 10 / Line 229-232 / Comment: I believe that the way in which the results are being presented can be standardized. At the beginning of the sentence, the authors talk about the variables that did not show a difference and chose to show the mean values of only one of them. Still at the end, t and p values of variables that showed difference are brought. Perhaps table 1 could contain this t and p information for all variables with no need to repeat in the text.

Author response:

With regards to the participant demographics results, we added p-values to Table 1, and removed numbers from the text of the manuscript so that information is not repeated.

Table 1 / Pg 10 / Line 235 / Comment: It seems contradictory to me to call the participants "NON-exercises" and say that they practice running, cycling, basketball and tennis. It may be necessary to revise the term in the text as a whole.

Author response:

We have revised the terms across the manuscript to call the groups MOVE (meets optimal volume of exercise) and notMOVE (does not meet optimal volume of exercise), to better reflect how we separated the groups by the guidelines.

Discussion / Pg 16 / Line 343-344 / Comment: The purpose of the study highlighted in the abstract, introduction and now in the discussion are presenting different objectives between them. I suggest reviewing.

Author response:

We have updated the purpose (in the abstract, introduction, and discussion) to state:

“The purpose of this study was to determine whether self-reported exercise behavior influences gait adaptation in young adults.”

Discussion / Pg 16 / Line 349-351 / Comment: The sentence: "young adults who habitually exercise have a higher tolerance for the energetically-challenging asymmetric belt speeds and adapt more gradually than THOSE WHO DO NOT", seems imprecise since according to table 1, all individuals practiced exercise. The difference between them is in the amount. This term "who do not exercise" appears at other times in the text, for example in line 376. I suggest reviewing the entire text.

Author response:

We have revised the terms across the manuscript to call the groups MOVE (meets optimal volume of exercise) and notMOVE (does not meet optimal volume of exercise), to better reflect that some participants in the notMOVE group do exercise, but just not up to the recommended amount.

Discussion / Pg 16 / Line 357-366 / Comment: I assume that all information provided in this excerpt is indeed relevant. However, it seems to me that a more adequate path could draw a parallel between the amount of physical activity practiced weekly and the potential influences that this generates on exercise capacity, VO2, gait, etc. Although what is written is true, the way it is, it implies that the participants of the NON-EXERCISE group do not practice any activity, which was not demonstrated by table 1. I suggest reviewing this and other excerpts that deal with the theme in this way , so that the differences between trained, insufficiently trained and untrained individuals can be clearer.

Author response:

First, we note that we have changed the group naming to “MOVE” and “notMOVE” to better reflect the difference between the two groups, as the reviewers pointed out that the “Non-exerciser” group did have some individuals who engaged in suboptimal amounts of weekly exercise. Given this update to the group names, we have updated the discussion to better reflect that response to the split-belt treadmill perturbation is likely a spectrum related to the amount of exercise a young adult engages in.

Line 369-376: “Our results suggest that young adults, regardless of exercise habits, can reduce positive work by the fast leg during split-belt adaptation, but that the amount of exercise practiced weekly and regularly affects the rate at which positive work rate reduction occurs. Those engaging in suboptimal amounts of exercise (notMOVE) may have a direr need to reduce positive work by the leg than those who engage in sufficient amounts of exercise (MOVE), because they may be working at a higher relative effort, resulting in a stronger incentive to reduce energetic and therefore mechanical work [15,17].”

Discussion / Pg 16-17 / Line 367-370 / Comment: Truncated sentence, I suggest rephrasing.

Author response:

We revised the sentence.

Line 377-380: “While adults who engage in more exercise per week do reach a minimum cost of transport during running, unlike those engaging in less exercise [28], our results suggest that adults who engage in more exercise per week do not have an advantage in reaching a minimum energetic cost during split-belt adaptation.”

Discussion / Pg 17 / Line 376-377 / Comment: I'm not sure this would be the best evidence for the moment. This paper (REF 33) talks about athletes and static balance.

Author response:

We updated the citation and the text to instead describe the differences between athletes and non-athletes in balance performance after a block of multimodal balance training.

Line 386-388: “Indeed, athletes perform better than non-athletes on a balance task after a learning block of multimodal balance training [39].”

Discussion / Pg 18 / Line 397-398 / Comment: Why not use the acronym SLA?

Author response:

We have updated the manuscript to ensure that acronyms and initialisms are used, where appropriate, including this sentence.

---

## [Decision Letter · Decision Letter 1]

9 May 2023

PONE-D-23-01404R1Habitual exercise evokes fast and persistent adaptation during split-belt walkingPLOS ONE

Dear Dr. Brinkerhoff,

Thank you for submitting your manuscript to PLOS ONE. After careful consideration, we feel that it has merit but does not fully meet PLOS ONE’s publication criteria as it currently stands. Therefore, we invite you to submit a revised version of the manuscript that addresses the points raised during the review process.

We look forward to receiving your revised manuscript.

Kind regards,

Flávio Oliveira Pires, PhD

Academic Editor

PLOS ONE

Journal Requirements:

Reviewers' comments:

Reviewer's Responses to Questions

**Comments to the Author**

1. If the authors have adequately addressed your comments raised in a previous round of review and you feel that this manuscript is now acceptable for publication, you may indicate that here to bypass the “Comments to the Author” section, enter your conflict of interest statement in the “Confidential to Editor” section, and submit your "Accept" recommendation.

Reviewer #1: (No Response)

Reviewer #2: All comments have been addressed

2. Is the manuscript technically sound, and do the data support the conclusions?

Reviewer #1: Yes

Reviewer #2: Yes

3. Has the statistical analysis been performed appropriately and rigorously? 

Reviewer #1: Yes

Reviewer #2: Yes

4. Have the authors made all data underlying the findings in their manuscript fully available?

Reviewer #1: (No Response)

Reviewer #2: Yes

5. Is the manuscript presented in an intelligible fashion and written in standard English?

Reviewer #1: Yes

Reviewer #2: Yes

6. Review Comments to the Author

Reviewer #1: This revised manuscript is much improved and in my opinion very close to being ready for publication. I only have one remaining minor point and congratulate the authors.

1. ll. 354-356: Points 1 and 2 of the results summary seem a bit at odds when it is not clearly stated that 1 (if I read it correctly) refers to the slow component. It would be useful to add this information.

Reviewer #2: General: The authors showed great care in reviewing the manuscript, presenting relevant resolutions for the comments referring to the first review. The document is already in conditions to be published, with the need for small and simple adjustments.

Introduction: Page 4, Line 88-90: I believe that the sentence "such that people with more aerobic training reach a minimum cost of transport while running, but those who engage in less aerobic training do not" could be improved.

Introduction: Page 4, Line 93-96: I'm not sure the word "conversely" (Line 95) is used correctly. After all, it seems to me that the two statements that the word connects are not in disagreement. In fact I understood that the second sentence only justifies the first, in which individuals who engage in more exercise achieve an optimal energy cost due to greater submaximal tolerance.

Discussion: Page 19, Line 425-432: I believe this passage is out of place. In my view part of it should be in the methods section and the other part in the results section.

7. PLOS authors have the option to publish the peer review history of their article (what does this mean?). If published, this will include your full peer review and any attached files.

Reviewer #1: No

Reviewer #2: No

---

## [Author Response · Author response to Decision Letter 1]

15 May 2023

Reviewer 1:

This revised manuscript is much improved and in my opinion very close to being ready for publication. I only have one remaining minor point and congratulate the authors.

Author response:

We thank the reviewers for their careful and insightful reviews that contributed to the improvement of the manuscript.

1. ll. 354-356: Points 1 and 2 of the results summary seem a bit at odds when it is not clearly stated that 1 (if I read it correctly) refers to the slow component. It would be useful to add this information.

Author response:

We clarified the main findings in the discussion summary as follows:

Line 355-358: “Our main findings are: 1) Compared to notMOVE, MOVE more gradually adapted the slow timescale of SLA, positive work rate by the fast leg, and negative work rate by the slow leg over the entirety of split-belt exposure, 2) MOVE initially adapted the fast timescale of SLA and positive work rate by the slow leg quicker than notMOVE…”

Reviewer 2:

General: The authors showed great care in reviewing the manuscript, presenting relevant resolutions for the comments referring to the first review. The document is already in conditions to be published, with the need for small and simple adjustments.

Author response:

We thank the reviewers for their careful and insightful reviews that contributed to the improvement of the manuscript.

Introduction: Page 4, Line 88-90: I believe that the sentence "such that people with more aerobic training reach a minimum cost of transport while running, but those who engage in less aerobic training do not" could be improved.

Author response:

We have split this sentence into two and revised it for clarity.

Line 87-90: “Moreover, the ability to achieve minimum energetic cost of transport while running is contingent upon a person’s level of aerobic training experience. People who engage in more aerobic training are able to reach this optimal cost of transport, whereas people with less aerobic training are not.”

Introduction: Page 4, Line 93-96: I'm not sure the word "conversely" (Line 95) is used correctly. After all, it seems to me that the two statements that the word connects are not in disagreement. In fact I understood that the second sentence only justifies the first, in which individuals who engage in more exercise achieve an optimal energy cost due to greater submaximal tolerance.

Author response:

The difference between the two statements is in the rate of adaptation due to exercise training – the first suggests that exercise-trained people adapted quicker, the second suggests that exercise-trained people adapt slower. These two hypotheses propose competing results, and we test which is more likely to be true in this study. We have revised this statement to hopefully read more clearly.

Line 93-98: “In this study, we examine two competing hypotheses regarding the adaptation to split-belt walking and its impact on energetic cost. The first hypothesis suggests that individuals who engage in more exercise would reach an energetic optimum faster, aiming to reduce energetic cost [28]. Conversely, the second hypothesis suggests that individuals who engage in more exercise may more gradually approach an energetic optimum due to their greater tolerance for submaximal exercise [31,32].”

Discussion: Page 19, Line 425-432: I believe this passage is out of place. In my view part of it should be in the methods section and the other part in the results section

Author response:

In response to the reviewer's comment regarding the placement of a particular section in our manuscript, we respectfully disagree with their suggestion. The mentioned section does not pertain to a specific question or hypothesis of the paper, but rather serves as a follow-up analysis aimed at determining the validity of a discussion point. As such, we believe it is more appropriately placed within the discussion section, where we critically analyze and interpret our findings in relation to the broader context of the study. We believe that including this analysis in the methods and results sections would disrupt the flow and coherence of our manuscript.

---

## [Editor Report · Decision Letter 2]

17 May 2023

PONE-D-23-01404R2Habitual exercise evokes fast and persistent adaptation during split-belt walkingPLOS ONE

Dear Dr. Brinkerhoff,

Thank you for submitting your manuscript to PLOS ONE. After careful consideration, we feel that it has merit but does not fully meet PLOS ONE’s publication criteria as it currently stands. Therefore, we invite you to submit a revised version of the manuscript that addresses the points raised during the review process.

I congratulate you by your study. Before final acceptance, please consider including the add-hoc analysis in results section, then you may briefly discuss these results and make the discussion a bit shorter. Thank you!

We look forward to receiving your revised manuscript.

Kind regards,

Flávio Oliveira Pires, PhD

Academic Editor

PLOS ONE

Journal Requirements:

Additional Editor Comments:

Dear Authors,

I congratulate you by your study. Before a final acceptance, please consider describing the add-hoc analysis in results section, then you may briefly discuss these results in the discussion section, making the discussion a bit shorter. Thank you!

---

## [Author Response · Author response to Decision Letter 2]

18 May 2023

Additional Editor Comments:

Dear Authors, I congratulate you by your study. Before a final acceptance, please consider describing the add-hoc analysis in results section, then you may briefly discuss these results in the discussion section, making the discussion a bit shorter. Thank you!

Author Response:

Thank you for your prompt review of our manuscript. We revised the post-hoc analysis so that the bulk of the information is in the results section (lines 307-314) and shortened the discussion section containing these results (lines 430-442).

---

## [Editor Report · Decision Letter 3]

22 May 2023

Habitual exercise evokes fast and persistent adaptation during split-belt walking

PONE-D-23-01404R3

Dear Dr. Brinkerhoff,

We’re pleased to inform you that your manuscript has been judged scientifically suitable for publication and will be formally accepted for publication once it meets all outstanding technical requirements.

Kind regards,

Flávio Oliveira Pires, PhD

Academic Editor

PLOS ONE
---

## [Editor Report · Acceptance letter]

24 May 2023

PONE-D-23-01404R3 

Habitual exercise evokes fast and persistent adaptation during split-belt walking 

Dear Dr. Brinkerhoff:

I'm pleased to inform you that your manuscript has been deemed suitable for publication in PLOS ONE. Congratulations! Your manuscript is now with our production department. 

Kind regards, 

on behalf of

BSc PhD Flávio Oliveira Pires 

Academic Editor

PLOS ONE